# Role of Prehabilitation and Rehabilitation on Functional Recovery and Quality of Life in Thyroid Cancer Patients: A Comprehensive Review

**DOI:** 10.3390/cancers15184502

**Published:** 2023-09-10

**Authors:** Lorenzo Lippi, Alessio Turco, Stefano Moalli, Marco Gallo, Claudio Curci, Antonio Maconi, Alessandro de Sire, Marco Invernizzi

**Affiliations:** 1Physical and Rehabilitative Medicine, Department of Health Sciences, University of Eastern Piedmont “A. Avogadro”, 28100 Novara, Italy; lorenzolippi.mt@gmail.com (L.L.); alessio.turco.phys@gmail.com (A.T.); stefano.moalli@libero.it (S.M.); 2Dipartimento Attività Integrate Ricerca e Innovazione (DAIRI), Translational Medicine, Azienda Ospedaliera SS. Antonio e Biagio e Cesare Arrigo, 15121 Alessandria, Italy; amaconi@ospedale.al.it; 3Endocrinology and Metabolic Diseases Unit, Azienda Ospedaliera SS. Antonio e Biagio e Cesare Arrigo, 15121 Alessandria, Italy; marco.gallo@ospedale.al.it; 4Physical Medicine and Rehabilitation Unit, Department of Neurosciences, ASST Carlo Poma, 46100 Mantova, Italy; claudio.curci@asst-mantova.it; 5Physical and Rehabilitative Medicine Unit, Department of Medical and Surgical Sciences, University of Catanzaro “Magna Graecia”, Viale Europa, 88100 Catanzaro, Italy; alessandro.desire@unicz.it; 6Research Center on Musculoskeletal Health, MusculoSkeletalHealth@UMG, University of Catanzaro “Magna Graecia”, 88100 Catanzaro, Italy

**Keywords:** thyroid cancer, rehabilitation, physical exercise, neck disorders, quality of life, fatigue, prehabilitation, telerehabilitation

## Abstract

**Simple Summary:**

This narrative review provides a comprehensive overview of the prehabilitation and rehabilitation strategies to manage thyroid cancer survivors, focusing on optimizing functional outcomes and enhancing their quality of life. The review highlights the importance of physical exercise in the rehabilitation process to improve cardiovascular fitness, muscle strength, and body composition, as well as to reduce fatigue. Scar management techniques, including soft tissue mobilization, silicone sheets, kinesiotaping, and laser therapy, can enhance functional recovery and cosmetic outcomes. Moreover, addressing dysphonia and dysphagia is crucial for improving the overall quality of life of these patients. Despite several barriers which still affect this multimodal rehabilitative approach, digital innovation might be an effective and sustainable tool with which to implement patient-centered management into the clinical setting. This review provides valuable insights into the current prehabilitation and rehabilitation strategies for thyroid cancer survivors, covering physical and psychological needs to optimize the functional outcomes and enhance the quality of life of these patients.

**Abstract:**

Background: This narrative review aims to provide a comprehensive overview of the current prehabilitation and rehabilitation strategies for thyroid cancer survivors to optimize functional outcomes and enhance their quality of life. Methods: The review follows the SANRA quality criteria and includes an extensive literature search conducted in PubMed/Medline, Web of Science, and Scopus. Results: The review emphasizes the role of a comprehensive rehabilitation approach in targeting the different domains that generate disability in thyroid cancer patients. In this context, physical activity, range of motion exercises, myofascial release, joint mobilization, and postural exercises are crucial for improving functional outcomes and reducing treatment-related discomfort and disability. Moreover, tailored rehabilitative management addressing dysphonia and dysphagia might have a positive impact on the quality of life of these patients. Despite these considerations, several barriers still affect the implementation of a multimodal rehabilitative approach in common clinical practice. Thus, sustainable and effective strategies like digital innovation and patient-centered approaches are strongly needed in order to implement the rehabilitative treatment framework of these subjects. Conclusions: This narrative review provides valuable insights into the current prehabilitation and rehabilitation strategies to treat thyroid cancer survivors, addressing physical, psychological, and vocational needs to optimize functional outcomes and enhance their quality of life.

## 1. Introduction

Thyroid cancer is the most common cancer of the endocrine system, with increasing incidence rates worldwide [1,2]. This highly prevalent cancer includes different types and subtypes (including papillary thyroid carcinoma, follicular thyroid carcinoma, medullary thyroid carcinoma, poorly differentiated thyroid carcinoma, and anaplastic thyroid carcinoma) [3]. Different subtypes have different characteristics and treatment approaches. The treatment of thyroid cancer often involves surgery, radioactive iodine therapy, and—in more advanced cases—second-line options such as radiotherapy, targeted therapy, immunotherapy, or (rarely) chemotherapy [4,5]. While the prognosis for thyroid cancer is generally excellent, patients with advanced disease (e.g., differentiated thyroid cancer progressively developing radioiodine-refractoriness) or with more aggressive subtypes (such as anaplastic thyroid cancer) show poorer survival rates and pose particular challenges regarding their management [6]. Moreover, the treatment process can exert a significant impact on various aspects of patient’s lives, including physical function, psychological well-being, and vocational function [7]. As a consequence, unsurprisingly, the recent systematic review by Walshaw et al. [8] underlined the growing research focusing on tailored interventions and addressing long-term consequences on health-related quality of life (HR-QoL) in thyroid cancer survivors.

In this context, rehabilitation strategies and physical exercise might play a crucial role in addressing these challenges, promoting optimal functional recovery and improving HR-QoL in cancer survivors [9,10,11]. Interestingly, exercise interventions have been shown to modulate immune function, reduce systemic inflammation, and enhance antioxidant defenses, contributing to improved overall health [10,12]. Physical therapy interventions targeting specific impairments can promote musculoskeletal strength, flexibility, and functional capacity [13], while comprehensive interventions can positively impact hormonal regulation and DNA repair mechanisms [14,15]. Thus, biological effects of rehabilitation interventions may enhance functional recovery, reduce treatment-related side effects, and improve long-term outcomes for cancer survivors [10,11,16,17].

Despite the growing clinical interest in the functional and psychological consequences of thyroid cancer [7,18], there is still a large gap in our knowledge regarding the optimal rehabilitation strategies to overcome the burden in terms of disability and decreased HR-Qol affecting thyroid cancer survivors. While different interventions have been proposed, the evidence of their effectiveness is limited, and cancer-specific strategies often lack consensus [13]. Furthermore, most of the studies focusing on rehabilitation interventions in thyroid cancer survivors have primarily examined specific aspects such as physical exercise or voice therapy, rather than providing a comprehensive rehabilitation strategy [19,20,21]. Taken together, these findings suggest the need for a comprehensive review summarizing the existing literature to better understand the most effective rehabilitation approaches to treat survivorship issues in thyroid cancer survivors, as well as prehabilitation strategies which can potentially help in dealing with the specific challenges posed by advanced disease management.

Thus, the aim of this narrative review was to provide a comprehensive overview of the current rehabilitation and prehabilitation strategies addressing the physical, psychological, and vocational needs of thyroid cancer survivors in order to optimize functional outcomes, reduce disability, and enhance the HR-Qol of these patients.

## 2. Research Methodology

This narrative review was designed following the SANRA quality criteria [22]. Extensive literature searches were conducted on PubMed/Medline, Web of Science (WoS), and Scopus using Mesh terms such as “Thyroid Cancer,” “Exercise”, “Prehabilitation”, “Rehabilitation”, “Physical Activity”, “Voice Therapy”, “Function”, “Performance”, “Disability”, and “Quality of Life”. The SPIDER tool search strategy [23] is summarized in Table 1.

Between January 2023 and March 2023, two independent reviewers (L.L. and A.T.) conducted the literature search. Subsequently, the identified studies were screened for eligibility by the reviewers. If a consensus could not be reached, a third reviewer (M.I.) was consulted. The inclusion criteria encompassed studies that addressed the research question of “What are the optimal prehabilitation and rehabilitation strategies to address the complex disability of patients with thyroid cancer?”. In detail, eligible articles involved human subjects with thyroid cancer, investigated the multidimensional aspects of disability in thyroid cancer survivors, and evaluated the effectiveness of rehabilitation interventions in improving functional outcomes, quality of life, and psychosocial well-being in this population. Studies in languages other than English; studies without full-text availability; studies not involving humans; and conference abstracts and master’s or doctoral theses were excluded.

Qualitative methods were used for data extraction and synthesis. The reviewers (L.L. and A.T.) independently extracted and synthesized information on the different prehabilitation and rehabilitation strategies for patients with thyroid cancer. In cases of disagreement, a third reviewer (M.I.) provided input. Considering the heterogeneity of the included studies and the narrative review’s design, a qualitative synthesis approach was used, presenting all outcome data in a narrative manner.

## 3. Physical Exercise and Rehabilitation

The active treatments targeting thyroid cancer might have a significant impact on the physical functioning of cancer survivors [24]. Recent research has underlined that surgical interventions, including thyroidectomy, might result in muscle weakness, reduced cervical range of motion, and postoperative pain [25,26]. Moreover, radioactive iodine therapy or external beam radiation, as well as multikinase inhibitor therapy, may cause fatigue and a significant decrease in overall physical fitness levels [27,28]. As a result, physical exercise and rehabilitation strategies are promising treatments to address functional and physical performance issues in patients undergoing curative local treatments. Figure 1 summarizes different therapeutic interventions in patients with thyroid cancer undergoing curative local treatments.

### 3.1. Role of Physical Exercise in the Rehabilitation of Thyroid Cancer Survivors

Physical exercise plays a crucial role in the rehabilitation of thyroid cancer survivors, improving physical functioning, reducing treatment-related side effects, and enhancing the overall HR-QoL of these patients [13]. Exercise programs tailored to the specific needs and capabilities of thyroid cancer survivors have shown promising results [19,20,29,30], and different exercise modalities have been proposed so far.

Aerobic exercise, such as walking, jogging, or cycling, focuses on improving cardiovascular fitness and endurance. Interestingly, promising results have been reported to enhance energy levels, reduce fatigue, and improve overall physical functioning in cancer survivors [31]. In addition, previous studies have shown that regular aerobic exercise can positively impact body composition, reducing the risk of cardiovascular disease and promoting psychological well-being [32,33].

Concurrently, strength training exercises, including resistance or weight-bearing activities, aim to improve muscle strength and function. These exercises might help to counteract the muscle weakness and loss of muscle mass commonly observed in cancer survivors [34]. Moreover, strength training programs have been shown to enhance muscle strength, functional capacity, and bone health in cancer survivors, with positive implications in terms of physical functioning and HR-QoL [35,36,37].

On the other hand, flexibility exercises focus on improving joint mobility and range of motion. These exercises can be particularly beneficial for thyroid cancer survivors who experience postoperative stiffness or limited neck mobility [38]. Stretching exercises, yoga, and tai chi have all been shown to enhance flexibility, reduce muscle tension, and improve the overall physical well-being of cancer survivors, and might be considered in the long-term management of thyroid cancer patients [31,39,40].

### 3.2. Considerations for Designing Exercise Programs

Different studies have demonstrated the positive impact of physical exercise on several outcomes in thyroid cancer survivors. These include improvements in physical fitness, fatigue reduction, and enhanced overall quality of life [19,20,41]. Additionally, physical exercise has been associated with lower recurrence rates, decreased mortality, and improved long-term survival outcomes in cancer survivors [42,43].

When designing exercise programs for thyroid cancer survivors, it is essential to consider individual characteristics, treatment-related side effects, and functional limitations [44]. Figure 2 shows, in detail, the different therapeutic exercise interventions in thyroid cancer survivors.

The collaboration between different healthcare professionals, including oncologists, neck surgery specialists, endocrinologists, physical and rehabilitation medicine specialists, and physical therapists, can help to set up tailored exercise programs and ensure their safety and effectiveness [45]. Tailoring exercise progressions, proper warm-up and cool-down routines, and individualized modifications are crucial to improving personalized approaches to cancer rehabilitation [46]. Future research should focus on setting up optimal exercise modalities, dosages, and long-term adherence to promote the most effective rehabilitation outcomes.

In conclusion, these findings suggest that aerobic exercise, resistance training, and combined exercise programs might improve cardiovascular fitness, muscle strength, body composition, and fatigue in thyroid cancer survivors. Healthcare professionals should consider incorporating individualized exercise programs as part of a comprehensive rehabilitation program to enhance the physical recovery and HR-QoL of thyroid cancer survivors.

## 4. Neck Disorders and Rehabilitation

Thyroid cancer and its active treatments might lead to several neck disorders and complications that affect the physical functioning and quality of life of patients with thyroid cancer who are undergoing curative local treatment [8]. Surgical procedures, radiation therapy, and other interventions (e.g., interventional radiology approaches) may result in pain, stiffness, limited range of motion, and other discomfort in the neck region [38].

Thus, a comprehensive rehabilitation program should focus on the specific neck disorders and complications that can arise from active treatments for thyroid cancer, including range of motion exercises and physical therapy aimed at improving neck function and reducing the discomfort that frequently characterizes thyroid cancer survivors after surgery [8,47,48].

### Rehabilitation Strategies for Neck Disorders

After thyroid cancer surgery, patients may experience pain and stiffness in the neck area. The surgical incision and tissue trauma can lead to discomfort and limited mobility, affecting activities of daily living. Scar tissue formation is a common complication following thyroid surgery [8]. Scar tissue can develop around the incision site, causing tightness and restricting neck movement. Adhesions may form, leading to further limitations in range of motion [49]. Moreover, albeit rarely used, external beam radiation therapy can result in fibrosis and stiffening of the neck tissues [50]. This can lead to reduced flexibility and difficulty in performing neck movements.

In this context, range of motion exercises are essential in order to improve neck mobility and flexibility [47,48]. These exercises aim to stretch and strengthen the muscles and tissues in the neck region, promoting a wider range of movement and reducing stiffness. Various techniques, including active and passive exercises, may be employed to target specific neck movements [51]. In addition, techniques such as myofascial release and joint mobilization can help with alleviating discomfort and improving the neck range of motion [25].

Lastly, the correction of postural imbalances and the promotion of proper alignment can improve neck function and reduce patients’ discomfort [52]. Postural exercises, ergonomic modifications, and postural education throughout daily activities are crucial components of neck rehabilitation [52].

## 5. Surgical Scar Rehabilitation in Thyroid Cancer Survivors

Surgical scars are a common sequela of thyroid cancer treatment and can have a significant physical and psychological impact on these patients [53,54]. The visibility and texture of scars can affect body image, social participation, and overall well-being. Thus, effective scar management strategies are crucial in order to address both physical and psychological implications in these subjects [55], and different interventions might be effectively integrated into a comprehensive rehabilitation approach.

In particular, soft tissue mobilization techniques might be performed by applying gentle pressure to the scar tissue to improve circulation, soften the scar, and reduce tightness [56,57]. Moreover, silicone sheets are commonly used in scar management to optimize scar healing [58]. These thin sheets create a barrier that helps to maintain moisture, reduce collagen production, and flatten and soften the scar tissue [59]. Silicone sheets can be applied directly to the scar and worn consistently for an extended period to achieve optimal results. Similarly, kinesiotaping can have positive implications for the rehabilitation treatment of surgical scars [60]. Scar tissue management may benefit from providing gentle and constant pressure, promoting scar alignment, and reducing scar adhesions [60]. Kinesiotaping may help to reduce pain, improve lymphatic drainage, provide postural support, and enhance patient awareness and education [9,61,62,63].

In this context, a biophysical approach including instrumental therapies might be considered. In more detail, laser therapy is an advanced technique commonly used in scar management for several disorders [55,64]. The laser energy targets the scar tissue to stimulate collagen production, promoting tissue remodeling and reducing scar visibility [65]. However, to date, some concerns still exist regarding the potential role of laser therapies in cancer patients. Other potential treatments, other than topical medication, in post-surgical scar management include prophylactic external beam radiation, microneedling, and dermabrasion and fillers, which might be considered based on the surgical scar’s characteristics [55,65].

Besides the body image implications and functional tissue retraction, tailored surgical scar pain management is a critical aspect of postoperative care following thyroid cancer surgery [66]. Pain at the incision site can significantly impact the patient’s comfort and quality of life and represents a crucial barrier to rehabilitation treatments. Various strategies, including pharmacological and non-pharmacological approaches, have been employed to address surgical scar pain [67]. In this scenario, studies have shown that scar massage has been associated with reduced pain and improved range of motion [68]. In accordance with this observation, laser therapy and kinesiotherapy might also play a role in the multimodal management of surgical scar pain [69].

Interestingly, botulinum toxin (Bont-A) has been proposed as a promising therapeutic option for the management of surgical scar pain in patients affected by thyroid cancer [70,71]. The neurotoxic properties of botulinum toxin allow for a targeted intervention at the pain site, providing localized pain relief and improving functional outcomes [72]. Bont-A acts on the release of neurotransmitters involved in pain signaling, such as substance P and glutamate, and can effectively reduce pain perception and alleviate the discomfort related to surgical scars [73]. Furthermore, its well-known muscle-relaxant effects may help to alleviate tension and spasms in the surrounding tissues, contributing to pain reduction [72]. Although further research is warranted in order to optimize the treatment protocols and establish their long-term efficacy, early evidence suggests that Bont-A could be a promising intervention in the comprehensive pain management approach for surgical scar pain in thyroid cancer patients [70,71]. Figure 3 summarizes the surgical scar management options in thyroid cancer survivors in detail.

Altogether, this evidence underlined that surgical scars resulting from thyroid cancer surgical treatment can have a detrimental physical and psychological impact on survivors. Effective rehabilitation strategies are crucial to addressing both scar healing and the functional consequences of thyroid surgery. Integrating these interventions into comprehensive rehabilitation programs can optimize scar healing, promote mobility, and improve the overall well-being of thyroid cancer survivors.

## 6. Dysphonia in Thyroid Cancer Survivors

Thyroid cancer and its active treatment can have a significant impact on vocal cords and voice quality, leading to dysphonia and voice-related impairments [74]. Surgical interventions, radiation therapy, and other treatment modalities may affect vocal cord function, resulting in changes in voice quality, pitch, loudness, and overall voice production [74,75]. Voice rehabilitation strategies, such as voice therapy and vocal exercises, play a crucial role in addressing dysphonia and improving vocal function in thyroid cancer survivors [76]. Figure 4 shows the potential rehabilitation strategies for the management of dysphonia in thyroid cancer survivors.

In particular, voice therapy is a specialized form of rehabilitation that focuses on improving vocal function and is mainly used for dysphonia treatment [77]. A trained speech–language therapist can guide individuals through different voice exercises and techniques to optimize vocal cord coordination, breath support, and resonance [78]. Moreover, speech–language therapists can perform vocal hygiene education and counseling for voice changes due to laryngeal physiology changes and surgical adhesions [76]. Similarly, laryngeal massage techniques might be effectively performed from the surgical site towards the surgical area in order to reduce tensions and improve vocal function [79]. Lastly, maintaining a proper posture might contribute to reducing muscle tension and could have positive implications on dysphonia rehabilitation [49].

Different studies have shown positive outcomes, including improvement in vocal cord function, vocal quality, and reduced voice-related distress, despite only moderate-quality evidence supporting voice therapy after thyroid surgery [76].

Vocal exercises are an integral part of voice rehabilitation for thyroid cancer survivors with post-surgery dysphonia [51]. These exercises aim to strengthen the vocal muscles, improve breath control, and promote optimal vocal cord vibration. They may include relaxation exercises, vocal warm-ups, pitch glides, resonance exercises, and articulation drills [78].

Lastly, educational therapy, including the maintenance of proper hydration and avoiding smoking and alcohol consumption, is a core component of a comprehensive rehabilitation approach to dysphonia [76], and avoiding throat clearing, shouting, and whispering are useful pieces of advice in the early phases after thyroid surgery [76].

Taken together, this evidence shows that thyroid cancer and its treatments can have a deep impact on vocal cords and voice quality, leading to detrimental consequences on psychosocial outcomes and the HR-QoL of these patients. Voice rehabilitation strategies offer effective interventions to address dysphonia and to improve vocal function in thyroid cancer survivors. The implementation of voice rehabilitation interventions into comprehensive rehabilitation programs for thyroid cancer survivors might optimize functional outcomes and improve the HR-QoL of thyroid cancer survivors.

## 7. Dysphagia in Thyroid Cancer Survivors

Thyroid cancer and its treatments can result in swallowing difficulties due to several complications, including intraoperative manipulation, retraction of scar tissue, cricopharyngeal dysfunctions, neural plexus injuries, strap muscle malfunction, and laryngotracheal fixation [80,81]. In addition, thyroid cancer treatments, such as surgery and radiation therapy, can result in pharyngeal dysfunction, affecting the swallowing mechanism [82]. Pharyngeal muscle weakness or impairment may lead to difficulties in coordinating swallowing movements, causing delays, aspiration, and choking during the swallowing process [83].

Similarly, esophageal dysfunction can also occur as a result of thyroid cancer treatment [82]. This may involve reduced esophageal motility, esophageal stricture formation, or gastroesophageal reflux, leading to difficulties in moving food through the esophagus and potential discomfort or pain during swallowing [80]. Moreover, dysphagia might be strongly related to both macronutrients and micronutrients deficits with potential implications in in skeletal muscle health of patients with cancer [84,85].

On the other hand, dysphagia in thyroid cancer survivors can lead to various complications, including malnutrition, dehydration, weight loss, infections, and respiratory issues [85,86]. Moreover, dysphagia significantly impacts the quality of life of thyroid cancer survivors, and specific strategies addressing this issue are needed [53,87].

Swallowing rehabilitation strategies, including swallowing exercises and dietary modifications, play a crucial role in managing dysphagia and improving swallowing function in thyroid cancer survivors [80].

In more detail, swallowing rehabilitation exercises aim to strengthen the swallowing muscles, improve coordination, and facilitate the swallowing process [88,89]. They may include tongue exercises, throat exercises, and swallowing maneuvers designed to enhance swallowing efficiency and reduce the risk of aspiration [82].

In addition, dietary modifications are crucial in managing dysphagia and ensuring safe and efficient swallowing. These modifications may involve altering food textures, such as transitioning to softer or pureed foods, and modifying liquid consistencies to reduce the risk of aspiration [90]. A multidisciplinary management involving speech–language pathologists and dietitians is essential in order to develop an individualized diet plan that meets the nutritional needs of advanced cancer patients and concurrently accommodates swallowing difficulties [85,91].

In addition to swallowing exercises and dietary modifications, the use of adaptive techniques and positioning can assist in managing dysphagia [92]. These techniques may include postural adjustments, such as chin tucks or head tilts, to optimize swallowing function and reduce the risk of aspiration [92]. Figure 5 summarizes dysphagia management strategies for thyroid cancer survivors.

In conclusion, dysphagia is a common sequela in thyroid cancer survivors due to its detrimental impact on functional recovery and HR-Qol. Swallowing rehabilitation strategies, including swallowing exercises, dietary modifications, and adaptive techniques, offer effective interventions to manage dysphagia and improve swallowing function. Integrating swallowing rehabilitation interventions into the comprehensive rehabilitation program might enhance functional recovery and promote synergisms between different rehabilitation strategies in thyroid cancer survivors.

## 8. Prehabilitation in Thyroid Cancer Patients

In recent years, growing attention has been paid to prehabilitation strategies for cancer patients. This approach involves multidisciplinary interventions aimed at improving the functional capacity of patients and minimizing the side effects associated with medical or surgical treatments. While traditional rehabilitation interventions are mainly focused on optimizing intra- and postoperative care, the preoperative period might be the most suitable time for cancer patients [93]. Although most patients with operable thyroid cancer are relatively young with few comorbidities, prehabilitation strategies play a crucial role in patients with advanced diseases [94]. In more detail, elderly patients with metastatic cancer are frequently affected by multiple comorbidities, with negative implications for systemic therapy side effects. As a result, a high prevalence of physical deconditioning and a risk of malnutrition were reported in these patients, highlighting the need for specific therapeutic interventions addressing this detrimental issue. In this context, managing modifiable risk factors before starting active cancer treatments might play a key role in the comprehensive management of thyroid cancer patients with advanced disease [93].

Interestingly, recent studies have shown that tailored prehabilitation programs might enhance cardiovascular endurance, muscular strength, and flexibility in cancer patients, with intriguing implications for tolerance of cancer treatments [95,96]. In particular, the exercise intervention is characterized by aerobic training, which can include activities such as walking; strength training; the use of free-weight or weight-bearing exercises; and flexibility exercises, such as stretching and relaxation techniques [96]. Additionally, prehabilitation interventions can help to maintain or improve functional capacity, enabling patients to perform daily activities and maintain their independence [97].

In more detail, patients undergoing lateral neck lymph node dissection might be characterized by disabling complications, including brachial plexus lesions and accessory nerve injuries [98]. Early detection of spinal accessory nerve and brachial plexus lesions should be mandatory for a tailored rehabilitation plan aiming at reducing functional limitations and disability. In this context, a detailed history and careful physical exam completed by EMG testing can help to guide clinicians in rehabilitation prescription and formulate prognoses [99]. In accessory nerve lesions, the upper trapezius loses its passive supportive function in scapular elevation, leading to scapulothoracic dysfunction when the arm is raised, causing the humeral head to lose its centrally located position within the glenoid fossa [100]. In accordance with this, a significant scapular dyssynergia might be underlined in patients with brachial plexus lesions, associated with whole-upper-limb functional impairment with significant implications for not only reaching function, but also grasping function [101]. As a result, upper limb function might be severely affected by this common complication, with a significant association between altered motor function, neck and shoulder pain, and disability [102].

In this context, both passive and active range of motion (ROM) exercises are essential for preventing shoulder complications, including adhesive capsulitis [103]. Moreover, other rehabilitation strategies might be considered, including exercise therapy, manual therapy, neuromuscular electrical stimulation, progressive strengthening exercises, and shoulder braces [103]. In this context, particular attention should be paid to a specific post-surgical rehabilitation approach, but also to a prehabilitation plan, before radiation therapies in order to prevent subsequent tissue fibrosis, inflammation, and worsening of shoulder function [103].

Another crucial benefit is the reduction in treatment-related side effects that represent a crucial burden of cancer treatments with detrimental consequences for the HR-QoL of cancer patients [93]. By optimizing patients’ nutritional statuses and overall health prior to treatment, prehabilitation can help to minimize the occurrence and severity of adverse effects associated with surgery, chemotherapy, targeted therapy, and radiation therapy [93,104].

Furthermore, prehabilitation programs often include psychological support and stress management techniques, addressing the emotional challenges that accompany a cancer diagnosis and treatment [105]. Lastly, prehabilitation has the potential to improve treatment outcomes by enhancing patients’ physical resilience and engagement in their own care [106]. In accordance with other cancer diseases, prehabilitation might be a suitable strategy by which to improve the functional outcomes and quality of life in thyroid cancer patients. In particular, the growing experience with targeted therapies for the treatment of advanced thyroid cancer and the increased confidence in the management of their adverse effects have increased the attention of clinicians to prehabilitation practices.

On the other hand, few studies focused on cancer-specific prehabilitation in thyroid cancer patients. More in detail, recent evidence suggests that prehabilitation might provide relevant and specific benefits in patients undergoing multikinase inhibitor treatments [107]. Moreover, a multimodal prehabilitation intervention might reduce the occurrence of side effects, including weight loss, anorexia, and hypertension, in thyroid cancer patients [108]. In more detail, the study by de Oca et al. [108] assessed the role of prehabilitation in radioactive iodine refractory differentiated thyroid cancer treated with lenvatinib. Multimodal prehabilitation intervention included medical optimization, physical intervention, and nutritional and psychological intervention. The exercise intervention was characterized by a home-based exercise program consisting of daily sessions of aerobic (walking, jogging, or cycling three times a week), strength (15 min daily of 8–10 repetitions for large muscles, by using patient-adapted weights or elastic bands and squats), and respiratory exercises. The nutritional intervention consisted of a personalized nutritional program designed for each patient, including food selection and meal planning. Despite the positive results, the low number of patients considered severely affects the clinical implications of the study results [108]. On the other hand, prehabilitation in thyroid cancer patients should also be focused on pre-existing comorbidities, tailoring a comprehensive intervention to the patient’s characteristics to enhance the patient’s functional capacity before cancer treatment. Figure 6 demonstrates the integration of prehabilitation in the comprehensive rehabilitation approach, aiming at enhancing functional recovery and quality of life of thyroid cancer patients with advanced disease.

Taken together, these findings highlight the potential benefits of prehabilitation in improving patients’ functional capacity, reducing treatment-related side effects, and enhancing overall well-being in individuals with thyroid cancer. Further research focusing on cancer-specific prehabilitation in this patient population is warranted in order to better understand its effectiveness and optimize its implementation in clinical practice.

## 9. Comprehensive Rehabilitation Programs: Barriers to and Strategies for Improving Patients’ Engagement

Comprehensive rehabilitation programs provide a holistic approach to cancer survivors’ care, targeting the multiple domains of physical and psychosocial functioning and enhancing the overall recovery and quality of life of these patients [13]. In order to simultaneously address the multiple aspects of functional recovery, a comprehensive rehabilitation program can lead to synergistic benefits and amplify the effects of other therapies [28,38,49,108].

Despite the significant advantages of a comprehensive rehabilitation approach, several barriers hinder the widespread implementation of rehabilitation programs in clinical practice.

Therefore, in the context of thyroid cancer rehabilitation, it is crucial to develop effective and sustainable strategies that improve patients’ engagement and prioritize patient-centered care [105]. Specific educational programs and a proper team of healthcare professionals are needed to increase the awareness of the impact of comprehensive rehabilitation interventions in thyroid cancer survivors [105].

Furthermore, the integration of emerging technologies and telemonitoring systems might provide additional benefits to both prehabilitation and rehabilitation interventions in cancer survivors [109,110]. These advancements aim to enhance a patient-centered approach to community care, empowering patients’ roles in functional recovery and offering sustainable solutions for targeting different health needs [110]. However, several concerns still affect digital implementation in the rehabilitation of thyroid cancer survivors, and a large gap in knowledge regarding the digital implementation in rehabilitation management of this specific population still persists.

To address these gaps, it is essential to incorporate telehealth education into healthcare professions’ curricula and ensure that all specialists have a transversal understanding of the basics of comprehensive rehabilitation programs and their relevance in the complex clinical management of cancer patients [111]. Interdisciplinarity is a crucial aspect in the complex framework of cancer patients’ care, and a better understanding of the different rehabilitation interventions can significantly improve the complex management of these patients and their outcomes [106].

On the other hand, digital implementation in rehabilitation might be a promising strategy for improving the sustainability of this therapeutic intervention. In thyroid cancer patients, careful economic consideration is necessary in order to maximize the clinical implementation of both prehabilitation and rehabilitation. In this context, research has underlined that the economic burden of thyroid cancer cases could surge to USD 3.5 billion by 2030 [112]. Thus, it should be noted that a precise stratification of patient characteristics should be mandatory in order to focus resources on patients with higher risks of complications and long-term functional impairment [113]. By incorporating comprehensive cost analyses, healthcare providers can make well-informed decisions tailored to patient demographics and available resources, enhancing the feasibility of these interventions [114]. Ongoing endeavors to refine and optimize cost-effective rehabilitation and prehabilitation strategies should be pursued, aiming to enhance their therapeutic impact while minimizing the economic burden [115,116].

In summary, the integration of comprehensive rehabilitation programs in thyroid cancer management requires efforts to enhance patients’ engagement and cost-effectiveness, and to increase health cancer professionals’ awareness and training. The availability of a specialized multidisciplinary team represents a crucial need for a holistic approach to thyroid cancer patients. Creating multidisciplinary and transdisciplinary teams composed of different professionals specializing in the management of thyroid cancer should be considered to enhance patient outcomes and minimize the functional implications of thyroid cancer treatments.

Although this comprehensive review presents study limitations, it aims to serve as a catalyst for future research focusing on the effects of comprehensive rehabilitation programs tailored to patients’ needs in order to reduce the multidomain disability of long-term cancer survivors. These interventions, though currently underestimated, could have the potential to significantly impact the HR-Qol of thyroid cancer survivors.

## 10. Conclusions

Taken together, the findings of the present comprehensive review demonstrate the need for the implementation of a specific and tailored rehabilitation plan to address the different disabling sequelae affecting thyroid cancer survivors, as well as patients undergoing targeted therapy for advanced radioiodine-refractory thyroid cancer. Indeed, a multimodal prehabilitative or rehabilitative approach has demonstrated significant benefits for thyroid cancer survivors, in terms of both functional outcome improvement and HR-QoL enhancement. In this scenario, emerging strategies, including prehabilitation and telerehabilitation, could be promising tools to enhance the continuum of care and sustainability of rehabilitation interventions, allowing thyroid cancer survivors to receive remote rehabilitation services and support in every phase of cancer treatment. Further research should focus on emphasizing the role of comprehensive rehabilitation programs in promoting long-term outcomes and sustainability for thyroid cancer survivors’ care.

## Figures and Tables

**Figure 1 cancers-15-04502-f001:**
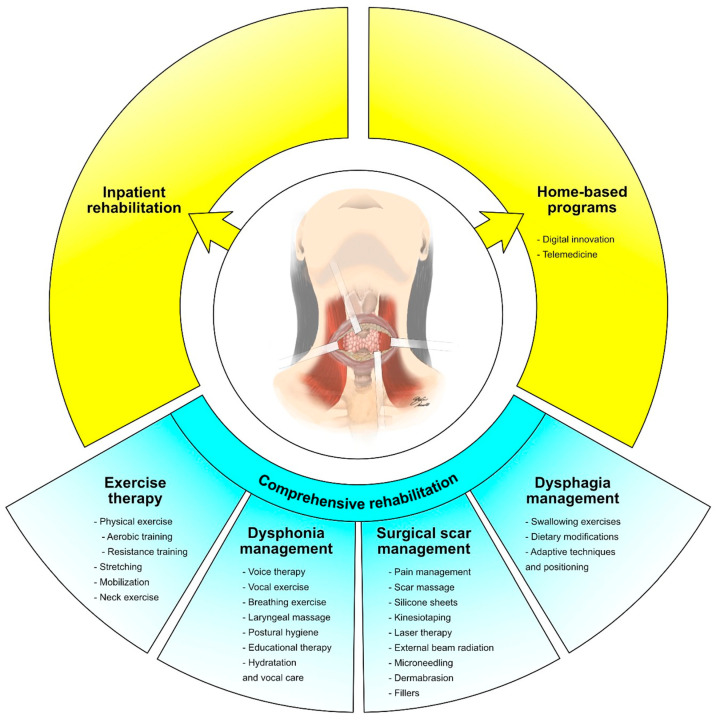
Comprehensive rehabilitation in patients with thyroid cancer undergoing curative local treatments.

**Figure 2 cancers-15-04502-f002:**
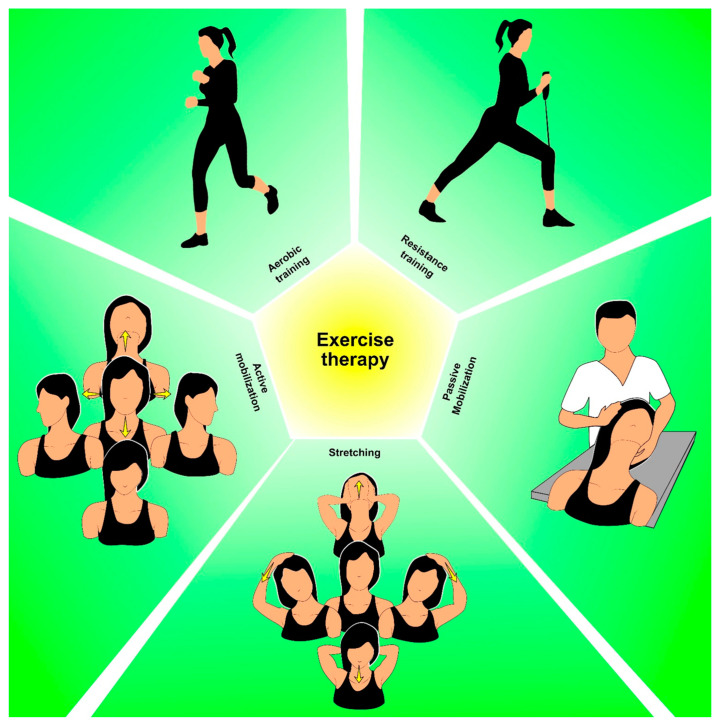
Therapeutic exercise intervention in thyroid cancer survivors.

**Figure 3 cancers-15-04502-f003:**
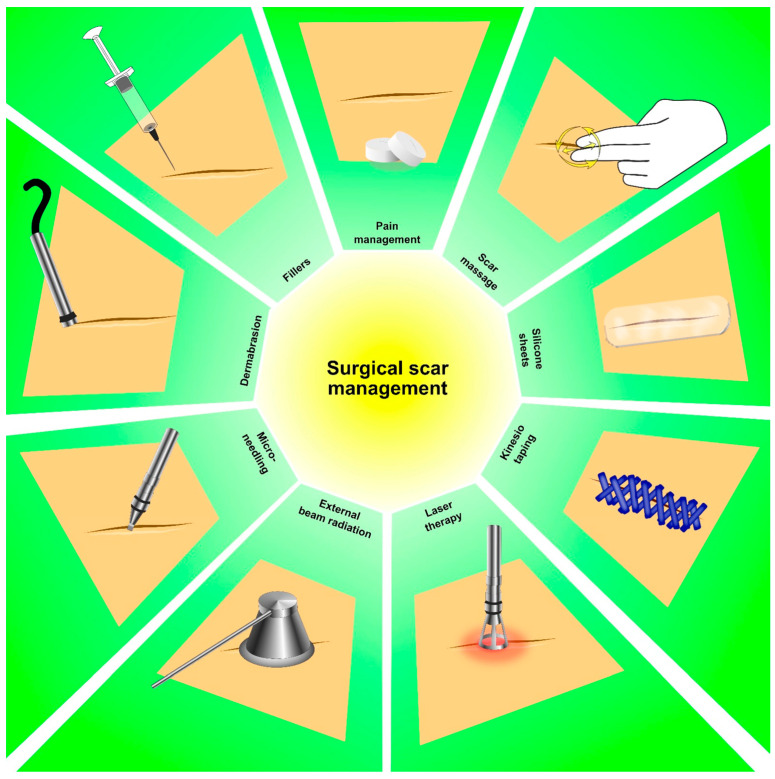
Surgical scar management options in thyroid cancer survivors.

**Figure 4 cancers-15-04502-f004:**
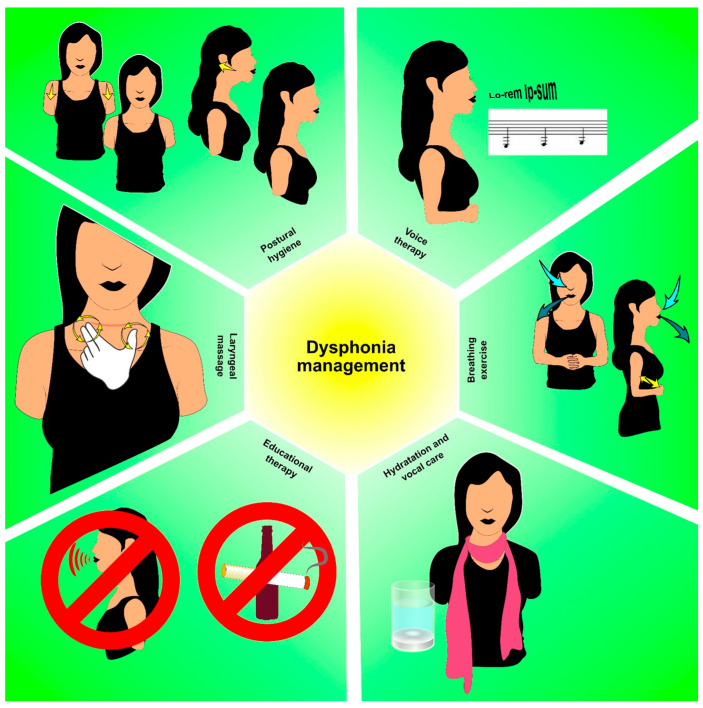
Rehabilitation strategies in thyroid cancer survivors with dysphonia.

**Figure 5 cancers-15-04502-f005:**
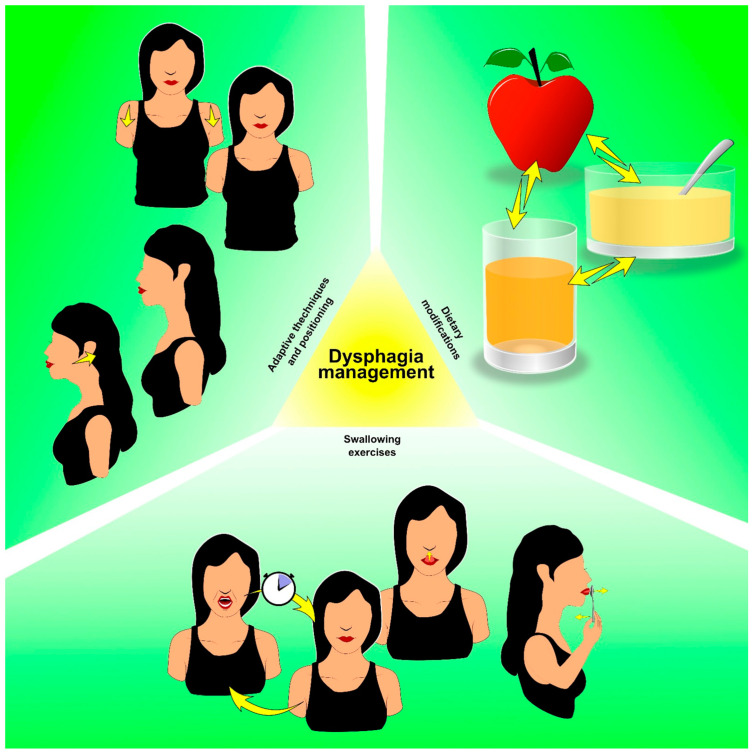
Rehabilitation strategies in thyroid cancer survivors with dysphagia.

**Figure 6 cancers-15-04502-f006:**
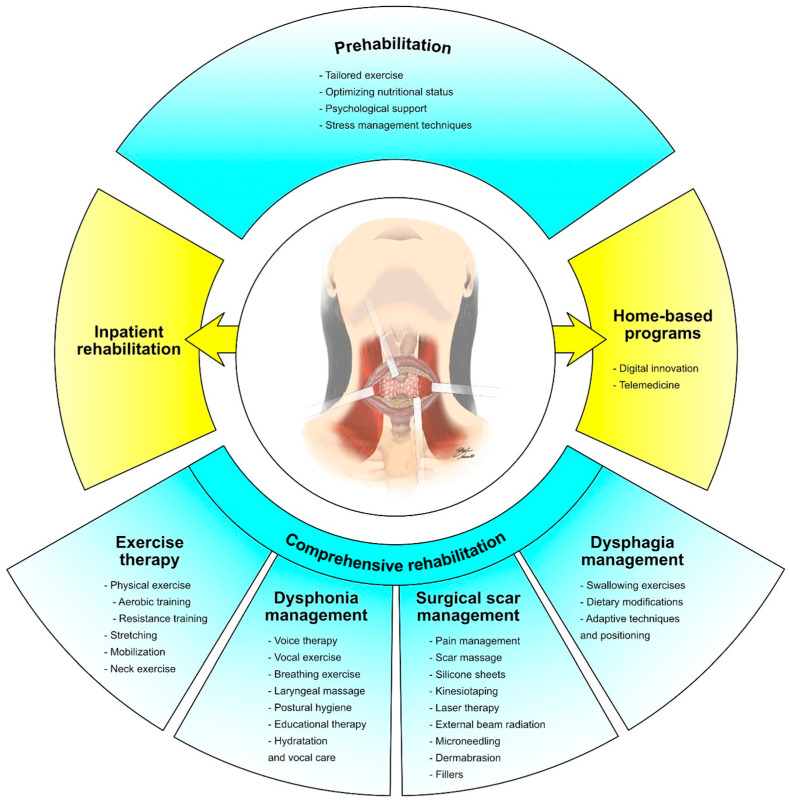
Comprehensive rehabilitation in thyroid cancer patients with advanced disease.

**Table 1 cancers-15-04502-t001:** Spider tool search strategy.

S	PI	D	E	R
Sample	Phenomenon of Interest	Design	Evaluation	Research Type
Patients with thyroid cancer	Rehabilitation Strategies	Any	Functional Outcomes	Qualitative
“Thyroid Cancer” “Neck Cancer”	“Exercise”“Prehabilitation”“Rehabilitation”“Physical Activity” “Voice Therapy”		“Function”“Performance”“Disability”, “Quality of Life”

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
