# Peer review of "Role of Prehabilitation and Rehabilitation on Functional Recovery and Quality of Life in Thyroid Cancer Patients: A Comprehensive Review"

_cancers, 2023, doi:10.3390/cancers15184502_

Round 1

Reviewer 1 Report

This is a helpful review of the evidence supporting rehabilitation strategies for patients experiencing local complications following treatment for thyroid cancer, and for prehabilitation strategies for patients being prepared for systemic treatment of advanced disease.

As a general point, I think that whilst both of these issues are very important, they relate to 2 very different groups of patients and are really quite separate. I think restructuring the paper to make a clearer distinction between the 2 issues would be helpful.

Within the introduction there appears to be a misunderstanding of the role of different treatments for thyroid cancer. At line 64, chemotherapy, radiotherapy, targeted therapies and immunotherapy are described as adjuvant therapies. An adjuvant therapy is a treatment given following surgery to improve the chance of cure. Of the treatments listed, only radiotherapy may be considered an adjuvant therapy. Chemotherapy is in fact very rarely used at all in modern thyroid cancer treatment, immunotherapy is not currently a licenced treatment for any type of thyroid cancer, and targeted treatments are palliative (ie non-curative, aimed at disease control/symptom relief) treatments. This should be clarified and is relevant to my point above about separating out different patient groups/treatment goals.

The research methodology is clearly described and appears to be appropriate, following well recognised criteria.

The following sections consider each of a number of relevant complications following thyroid cancer treatment, examining the evidence base for any therapeutic interventions. In general these sections are well structured and helpfully summarise the existing data. At line 286 I wonder whether the authors mean a speech and language THERAPIST rather than pathologist which is not a role description that I recognise.

In the section on prehabilitation, I think that it could be made clearer from the outset that in thyroid cancer this is really of greatest relevance to patients with advanced disease being considered for systemic therapy. The majority of operable thyroid cancer patients are relatively young with few co-morbidities and prehabilitation strategies that might be considered for example prior to surgery for lung or oesophageal cancer are rarely required.  Whilst mention is made of the evidence supporting prehabilitation programmes for example for patients due to start lenvatinib, it would be useful to include more detail in the text about exactly what interventions are considered beneficial along with any available quantitative data about the benefits.

Figure 1, whilst providing a helpful summary of rehabilitation strategies, again conflates the different patient groups- those having curative local treatment likely to require rehab and those with advanced disease requiring prehabilitation. I think it would be clearer if there were 2 figures clearly separating these groups. 

The section regarding barriers to providing a comprehensive rehabilitation programme is helpful, the most important point here being the availability of an appropriately skilled multidisciplinary team.

Overall I think this is a helpful review, but it would benefit from restructuring and some additional detail regarding prehabilitation strategies.

Minor adjustments required- eg consideration of changing 'speech and language pathologist' to 'speech and language therapist'.

Author Response

We would like to express our gratitude to the reviewers for their insightful comments and constructive feedback on our manuscript titled "Role of Prehabilitation and Rehabilitation on Functional Recovery and Quality of Life in Thyroid Cancer Patients: A Comprehensive Review." We appreciate the time and effort they dedicated to reviewing our work, and we have taken your suggestions into careful consideration to enhance the quality and clarity of our review.

Response to Reviewer 1

This is a helpful review of the evidence supporting rehabilitation strategies for patients experiencing local complications following treatment for thyroid cancer, and for prehabilitation strategies for patients being prepared for systemic treatment of advanced disease.

As a general point, I think that whilst both of these issues are very important, they relate to 2 very different groups of patients and are really quite separate. I think restructuring the paper to make a clearer distinction between the 2 issues would be helpful.

Thank you for your valuable comment. We appreciate your insights regarding the distinction between rehabilitation and prehabilitation. We have carefully improved the paper providing a clear differentiation between these two critical aspects. In particular, we better characterized that rehabilitation focused on patients undergoing curative local treatment and thyroids cancer survivors, while prehabilitation is specifically focused on addressing the needs of individuals with advanced diseases.

This restructuring will help ensure that the distinct nature of these two approaches is clearly conveyed, enhancing the clarity and relevance of the paper for different patient groups. We believe that this revision will contribute to a more coherent and focused presentation of our study's findings in accordance with the Reviewers’ concerns.

Within the introduction there appears to be a misunderstanding of the role of different treatments for thyroid cancer. At line 64, chemotherapy, radiotherapy, targeted therapies and immunotherapy are described as adjuvant therapies. An adjuvant therapy is a treatment given following surgery to improve the chance of cure. Of the treatments listed, only radiotherapy may be considered an adjuvant therapy. Chemotherapy is in fact very rarely used at all in modern thyroid cancer treatment, immunotherapy is not currently a licenced treatment for any type of thyroid cancer, and targeted treatments are palliative (ie non-curative, aimed at disease control/symptom relief) treatments. This should be clarified and is relevant to my point above about separating out different patient groups/treatment goals.

Thank you for your comment. We totally agree with the statement that RT, conventional chemotherapy, target therapy, and immunotherapy (which is increasingly used for the treatment of PL1/PD-L1 expressing anaplastic thyroid cancer and poor-differentiated thyroid cancer) can hardly be considered adjuvant therapies (ie, post-surgical therapies given to improve the chances of cure). To make our thinking clearer, we have replaced the term “adjuvant” with “second line options”, and reformulated the sentence accordingly:

"The treatment of thyroid cancer often involves surgery, radioactive iodine therapy, and -for more advanced cases- second line options such as radiotherapy, targeted therapy,  immunotherapy or (rarely) chemotherapy [4,5]". (Section “1. Introduction”)

The research methodology is clearly described and appears to be appropriate, following well recognised criteria.

We would like to thank the reviewer for the comment.

The following sections consider each of a number of relevant complications following thyroid cancer treatment, examining the evidence base for any therapeutic interventions. In general these sections are well structured and helpfully summarise the existing data. At line 286 I wonder whether the authors mean a speech and language THERAPIST rather than pathologist which is not a role description that I recognise.

We would like to sincerely thank for the helpful comment. We totally agree with the reviewer's observation, therefore the term has been corrected to the term "therapist". (Section “5. Dysphonia in Thyroid Cancer Survivors”)

In the section on prehabilitation, I think that it could be made clearer from the outset that in thyroid cancer this is really of greatest relevance to patients with advanced disease being considered for systemic therapy. The majority of operable thyroid cancer patients are relatively young with few co-morbidities and prehabilitation strategies that might be considered for example prior to surgery for lung or oesophageal cancer are rarely required.  Whilst mention is made of the evidence supporting prehabilitation programmes for example for patients due to start lenvatinib, it would be useful to include more detail in the text about exactly what interventions are considered beneficial along with any available quantitative data about the benefits.

We would like to thank the reviewer for the comment. We included further insight about patients considered eligible for pre-operative intervention and characteristics of prehabilitation programs. (Section “7. Prehabilitation in Thyroid Cancer Patients”)

Figure 1, whilst providing a helpful summary of rehabilitation strategies, again conflates the different patient groups- those having curative local treatment likely to require rehab and those with advanced disease requiring prehabilitation. I think it would be clearer if there were 2 figures clearly separating these groups. 

Thank you for your comment. We totally agree with your observation. Two separate figures have been created to provide more precise information and better characterize the interventions.

The section regarding barriers to providing a comprehensive rehabilitation programme is helpful, the most important point here being the availability of an appropriately skilled multidisciplinary team.

We would like to thank the reviewer for the comment. We've included further insights into the availability of specialized multidisciplinary teams. (Section “8. Comprehensive Rehabilitation Programs: Barriers and Strategies Improving Patients’ Engagement”)

Overall I think this is a helpful review, but it would benefit from restructuring and some additional detail regarding prehabilitation strategies.

We would like to express our gratitude to the reviewer for their insightful comments and constructive feedback. We have drastically improved the Prehabilitation Section (7. Prehabilitation in Thyroid Cancer Patients) in accordance with the Reviewer’s comment. We appreciate the time and effort they dedicated to reviewing our work.

Reviewer 2 Report

The present narrative review aims to provide an overview about the current rehabilitation and prehabilitation strategies addressing the physical, psychological, and vocational needs of thyroid cancer survivors in order to optimize functional outcomes, reduce disability and enhance the HR-QoL of these patients. This is a comprehensive review focusing on a specific issue that will increase attention in the future and for future research, even if the evidence is quite low so far.

I have only few comments:

  1. It should be underlined throughout the text that thyroid cancer has usually an indolent course, low morbidity and a good prognosis. Therefore, pre and rehabilitation programs should be reserved to high risk patients with advanced disease (that is a minority). It should be underlined in the text, explaning also the benefits of these programs and the costs.
  2. More figures should be added explaining the rehabilitation exercises for each functional disability.
  3. Thyroid cancer has a different treatment and course from others head and neck cancers; it should be underlined in the text (in some paragraphs they are discussed as the same entity but surgical and postsurgical treatments and prognosis are very different)
  4. I suggest to add a paragaph about the pre and rehabilitation programs in patients undergoing lateral neck lymph node dissection; brachial plexus lesions, accessory nerve injury are common complications after surgery for metastatic thyroid cancer. This is a very important issue for both clinicians and patients.

only minor editing of english language 

Author Response

We would like to express our gratitude to the reviewers for their insightful comments and constructive feedback on our manuscript titled "Role of Prehabilitation and Rehabilitation on Functional Recovery and Quality of Life in Thyroid Cancer Patients: A Comprehensive Review." We appreciate the time and effort they dedicated to reviewing our work, and we have taken your suggestions into careful consideration to enhance the quality and clarity of our review.

The present narrative review aims to provide an overview about the current rehabilitation and prehabilitation strategies addressing the physical, psychological, and vocational needs of thyroid cancer survivors in order to optimize functional outcomes, reduce disability and enhance the HR-QoL of these patients. This is a comprehensive review focusing on a specific issue that will increase attention in the future and for future research, even if the evidence is quite low so far.

I have only few comments:

It should be underlined throughout the text that thyroid cancer has usually an indolent course, low morbidity and a good prognosis. Therefore, pre and rehabilitation programs should be reserved to high risk patients with advanced disease (that is a minority). It should be underlined in the text, explaining also the benefits of these programs and the costs.

We would like to thank the reviewer for the comment.  We added an “economic considerations” paragraph following your suggestion. (Section “8. Comprehensive Rehabilitation Programs: Barriers and Strategies Improving Patients’ Engagement”). Regarding the information on high-risk patients with advanced disease, this information was provided in the Introduction section in accordance with the Reviewer’s comment.

More figures should be added explaining the rehabilitation exercises for each functional disability.

Thank you for your comment. Four different figures were provided further characterizing the rehabilitation interventions in patients with thyroid cancer.

Thyroid cancer has a different treatment and course from other head and neck cancers; it should be underlined in the text (in some paragraphs they are discussed as the same entity, but surgical and postsurgical treatments and prognosis are very different)

Thank you for your comment. We totally agree with your observation. In order to clarify this point, we have removed “Neck Cancer” from the text reporting information only from patients with thyroid cancer (Section “3. Neck Disorders and Rehabilitation”). Furthermore, we removed the references dealing exclusively with "head and neck cancer treatment", implementing the information regarding thyroid cancer patients.

I suggest to add a paragraph about the pre and rehabilitation programs in patients undergoing lateral neck lymph node dissection; brachial plexus lesions, accessory nerve injury are common complications after surgery for metastatic thyroid cancer. This is a very important issue for both clinicians and patients.

We would like to thank the reviewer for the comment.  We added in the “Prehabilitation in Thyroid Cancer Patients” a specific paragraph focusing on the role of pre and rehabilitation programs in patients undergoing lateral neck lymph node dissection, focusing on the brachial plexus lesions and accessory nerve injuries after surgery for metastatic thyroid cancer.

Round 2

Reviewer 1 Report

Thank you for addressing my previous comments. I do not have anything further to add.

The English in one or two of the added sections would benefit from review- eg at lines 518-519- 'health cancer professional awareness' is not very clear.

Author Response

We would like to express our gratitude to the reviewer for his insightful comments and constructive feedback on our manuscript titled "Role of Prehabilitation and Rehabilitation on Functional Recovery and Quality of Life in Thyroid Cancer Patients: A Comprehensive Review." We appreciate the time and effort you dedicated to reviewing our work.

Reviewer 2 Report

I think that the authors have adequately addressed the comments in the
revised version of the manuscript. However, I suggest to add a more detailed paragraph about the pre and rehabilitation programs in patients undergoing lateral neck lymph node dissection (brachial plexus lesions, accessory nerve injury).

I have no further comments.

 Minor editing of English language required

Author Response

We would like to express our gratitude to the reviewer for his insightful comment. We provided more details about the pre and rehabilitation programs in patients undergoing lateral neck lymph node dissection in accordance with the reviewer’s comment. Manuscript changes were highlighted in yellow in the text.